# Tuning the catalytic CO hydrogenation to straight- and long-chain aldehydes/alcohols and olefins/paraffins

Yizhi Xiang[1] & Norbert Kruse[1,2]

The catalytic CO hydrogenation is one of the most versatile large-scale chemical syntheses leading to variable chemical feedstock. While traditionally mainly methanol and long-chain hydrocarbons are produced by CO hydrogenation, here we show that the same reaction can be tuned to produce long-chain $n$-aldehydes, 1-alcohols and olefins, as well as $n$-paraffins over potassium-promoted CoMn catalysts. The sum selectivity of aldehydes and alcohols is usually $>50$ wt% whereof up to $\sim 97\%$ can be $n$-aldehydes. While the product slate contains $\sim 60\%$ $n$-aldehydes at $p_{H2}/p_{CO} = 0.5$, a 65/35% slate of paraffins/alcohols is obtained at a ratio of 9. A linear Anderson–Schulz–Flory behaviour, independent of the $p_{H2}/p_{CO}$ ratio, is found for the sum of $C_{4+}$ products. We advocate a synergistic interaction between a $Mn_5O_8$ oxide and a bulk $Co_2C$ phase, promoted by the presence of potassium, to be responsible for the unique product spectra in our studies.

[1] Voiland School of Chemical Engineering and Bioengineering, Washington State University, Pullman, Washington 99164, USA. [2] Institute for Integrated Catalysis, Pacific Northwest National Laboratory, Richland, Washington 99332, USA. Correspondence and requests for materials should be addressed to N.K. (email: norbert.kruse@wsu.edu).

In his seminar 'Zwölf Jahre Kohlenforschung' (Twelve Years of Carbon Research, 1926) at the Kaiser-Wilhelm-Institut für Kohlenforschung in Mülheim-Ruhr (today Max Planck Institute für Kohlenforschung)[1], Fischer informed the Kuratorium about his incapacity of reproducing the 1913 patent by Badische Anilin and Soda Fabriken (BASF) claiming hydrocarbon chain initiation through CO hydrogenation to occur in an excess of carbon monoxide[2]. On the same occasion, Fischer reported that an excess of hydrogen was actually necessary for hydrocarbon chains to form. He was puzzled by the observation that, contrary to the BASF patent claim, the oily product fraction contained no hydrocarbons but oxygenates instead. In early pilot tests, this product fraction, which he coined 'synthol', turned out to have physical properties enabling its use as a transportation fuel.

Mittasch (leading the research at BASF)[3] as well as Fischer and Tropsch[4,5] chose a potassium-promoted iron catalyst for their experiments. Although in the following decades >20,000 different catalysts were empirically screened, potassium-promoted supported iron catalysts stood out for their low-cost production and good performance until present days. Similarly, following Fischer's original suggestion, process conditions were adjusted to establish $H_2$/CO pressure ratios between 1 and 2, with the actual value mainly depending on the extent of the catalyst's water gas shift activity (producing additional hydrogen according to $CO + H_2O \leftrightarrow CO_2 + H_2$) and on the product classes envisaged in the synthesis reaction. Decades after the original discoveries by Mittasch, and Fischer and Tropsch, the influence of the $H_2$/CO pressure on the synthesis kinetics for potassium-promoted iron catalysts was systematically investigated by Dry et al.[6], and Matsumoto and Satterfield[7]. These authors established a first-order kinetic dependence on hydrogen and a hydrocarbon chain-lengthening probability insensitive to both the actual $H_2$/CO ratio and the degree of potassium promotion. It seems that similar studies for other catalysts enabling paraffins, olefins and oxygenates formation were not conducted up to date. This is all the more surprising against the background that catalyst formulations with specific selectivities for these product classes were tailored in the more recent past[8–12].

Based on our previous investigations of the catalytic CO hydrogenation to long-chain alcohols over ternary CoCuMn catalysts, we demonstrate here the very strong influence of the $H_2$/CO pressure ratio on the product spectrum. In particular, we provide results for a potassium-promoted CoMn ('CoMnK' or CoMnK for simplicity, to indicate the general composition) catalyst producing straight- and long-chain aldehydes, terminal alcohols and olefins, as well as n-paraffins while changing the $H_2$/CO ratio from low to high, at otherwise constant chain-lengthening probability. While aldehydes are presently produced via homogeneous hydroformylation of terminal olefins on an industrial scale, with the possibility of reducing them to terminal alcohols, our results open the door for designing a one-pot heterogeneous process with varying $H_2$/CO ratios to optimize the selectivities of either product class.

## Results

**Catalytic data.** The rationale behind using alkali-promoted CoMn was twofold. On one hand, alkali is generally being considered a structural promoter influencing the product selectivity and increasing the overall rates of Co-based catalysts[13]. On the other hand, mixed-metal Co–Mn catalysts were previously reported to produce olefins with high selectivity[14–17]. To our surprise, CoMn prepared via oxalate precursors (see Methods), either alkali-promoted or not, turned out to produce significant amounts

of straight-chain oxygenates in terms of aldehydes and alcohols, besides olefins and paraffins. Supplementary Table 1 provides an overview of the selectivities obtained for binary CoMn catalysts. It is seen that up to ~25% oxygenates are obtained, with little influence of the actual Co/Mn ratio. Interestingly, >75% of the oxygenate fraction are n-aldehydes. For comparison, ternary CoCuMn catalysts were found to produce up to 60% alcohols under similar reaction conditions, with insignificant aldehyde side fractions[18,19].

The occurrence of considerable aldehyde selectivity with binary CoMn catalysts prompted us to tune their performance by including alkali in the catalyst preparation. Catalysts with a Co/Mn atomic ratio of 2/1 and 4/1 were chosen for this purpose since both demonstrated (slightly) higher aldehyde versus alcohol selectivity than those with a smaller ratio. The results in Fig. 1 refer to a $H_2$/CO pressure ratio of 1.5 and demonstrate the temperature dependence of product selectivities for CoMnK catalysts containing varying amounts of potassium in comparison with binary CoMn.

It is seen that all catalysts produce hydrocarbons, aldehydes and alcohols over the investigated range of temperatures. Oxygenates form with highest selectivities (calculations always include $CO_2$ production) under conditions of low CO conversion (low temperature). If the $CO_2$ production in our selectivity calculations is ignored (this procedure can be frequently found in the Fischer-Tropsch (F-T) literature with $H_2$-lean syngas), oxygenates selectivities up to ~50 wt% can be obtained for a CO conversion of ~40 % at high temperature (Supplementary Fig. 1). Potassium-containing catalysts clearly outperform binary CoMn in this low-temperature range. It also seems that catalysts with a lower amount of K ($Co_2Mn_1K_{0.1}$ and $Co_4Mn_1K_{0.1}$ where indices provide atomic ratios) are preferable over those with a

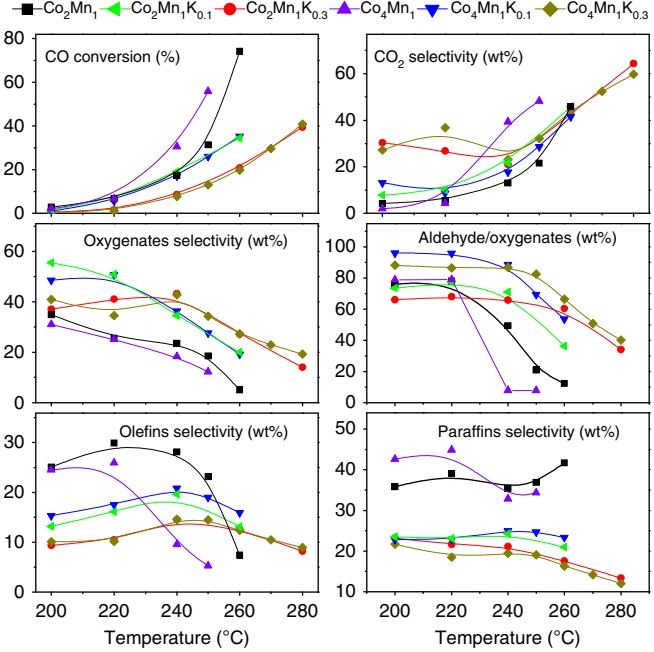

**Figure 1 | Catalytic performance.** Activity and selectivity of $Co_2Mn_1K_x$ and $Co_4Mn_1K_x$ (x = 0, 0.1 and 0.3) catalysts for CO hydrogenation. Catalytic tests were carried out at P = 40 bar, $H_2$/CO = 1.5 and total gas flow = 40 ml min$^{-1}$. The same batch of catalyst was subjected to conversion at different temperatures (sequential increase from low to high). The catalytic performance of $Co_4Mn_1K_{0.1}$ at 220 °C was reproduced at least five times with errors <10% over different batches of catalyst. The error for the total carbon balance was ~8% from three different measurements.

higher amount of K, if oxygenate production is the target of the performance optimization. This is particularly clear with regard to $CO_2$ production which, quite generally, increases with increasing temperatures (eventually reaching unacceptable levels at the highest temperatures). For all samples, the CO conversion increases exponentially with increasing temperature. The presence of potassium tends to decrease the conversion. This decrease is less pronounced for catalysts containing low amounts of K, so a compromise can be reached for the selectivity-conversion tradeoff by adjusting medium temperatures between 220 and 240 °C. For example, combined oxygenate/olefin selectivities of ~65% along with a CO conversion up to ~20% are obtained for $Co_4Mn_1K_{0.1}$ and $Co_2Mn_1K_{0.1}$ catalysts.

Based on the results presented so far, one might wonder how other alkali metals would influence the catalytic performance. To provide answers to this question, catalytic test studies were performed with $Co_4Mn_1Me_{0.1}$ (Me = Li, Na and K), see Supplementary Figs 2 and 3. It seems clear from these studies that potassium-promoted catalysts perform best in terms of oxygenate production. On the other hand, Li-containing catalysts increase the CO conversion at low temperatures along with a shift from oxygenate to olefin production. While this performance shift is interesting in itself, we concentrate in the following on the possibility of producing oxygenates with high selectivities using $Co_4Mn_1K_{0.1}$. In particular, we shall inspect the highly unusual capacity of this catalyst to produce oxygenates with a dominating fraction of aldehydes. Remarkably, according to Fig. 1, all CoMn catalysts, either in the presence or absence of potassium, produce oxygenates with a fraction of at least 65% aldehydes at low temperatures. The highest fractional selectivity $S_{RCHO/RO}$ of ~97 wt% is obtained for $Co_4Mn_1K_{0.1}$. For all potassium-containing catalysts, $S_{RCHO/RO}$ decreases significantly at temperatures beyond 240 °C. It seems that the presence of potassium somewhat widens the low-temperature window of high $S_{RCHO/RO}$ levels.

Another intriguing observation of the present study is made in relation to the carbon number distribution of aldehydes (or total oxygenates). As shown in Fig. 2, and in agreement with the results of Fig. 1, the aldehyde fraction of oxygenates is dominating whatever the hydrocarbon chain length, while alcohols play an insignificant role. The corresponding Anderson–Schulz–Flory (ASF) plots for aldehydes reveals major (negative) deviations for short-chain $C_1$ and $C_3$ aldehydes. While the absence of

formaldehyde would have been expected, that of the propionaldehyde is surprising in view of the large quantities produced for the preceding acetaldehyde homologue. Then, from $C_4$ onward, polymerization-type kinetics resulting in a strictly linear ASF behaviour is observed. The α value calculated for $C_{4+}$ oxygenates can be adjusted to between 0.4 and 0.7 depending on the relative atomic amounts of metals in CoMnK and the chosen temperatures (Supplementary Fig. 4). Obviously, a characteristic ASF distribution can be obtained for all CoMnK catalysts and allows an optimization of $C_{4+}$ long-chain or $C_2$–$C_4$ short-chain oxygenates production. (Supplementary Fig. 5 shows $C_{4+}$ fractions above 85% for $Co_2Mn_1K_{0.1}$ and $Co_4Mn_1K_{0.1}$ catalysts at low temperature, while the $C_2$–$C_4$ short-chain products are optimized for $Co_2Mn_1K_{0.3}$ and $Co_4Mn_1K_{0.3}$ catalysts at high temperature). According to Fig. 2, olefins and paraffins also show an ASF distribution with a chain-lengthening probability slightly larger than that of the aldehydes. A strictly linear behaviour is observed up from $C_3$. We also note that only terminal functionalization of the hydrocarbons is observed, that is, end-on oxygenates and olefins occur with no indication for branching of the hydrocarbon scaffold. The selectivity for $CH_4$ formation is favourably low for CoMnK catalysts; in the case of $Co_4Mn_1K_{0.1}$, it is as low as ~6 wt%.

The catalytic performance of CoMnK as shown above demonstrates the versatility of the F–T technology in providing various chemical feedstocks. A unique observation is made when subjecting a $Co_4Mn_1K_{0.1}$ catalyst to varying $H_2$/CO pressure ratios (Fig. 3). While aldehydes are clearly dominating at low $H_2$/CO ratios, they are insignificant when these ratios increase to high values. On the other hand, the alcohol fraction is strongly increasing at the expense of aldehydes and reaches a maximum at $H_2$/CO = 5. Paraffins increase from initially 20% and dominate for $H_2$/CO > 5, while olefins starting from a near to 20% level disappear at such high partial pressure ratios. The product spectrum at $H_2$/CO = 9 is made up of 65% (straight-chain) hydrocarbons and 35% terminal alcohols; no other products (including $CO_2$) are observed. It is remarkable that the CO conversion reaches 75% under these conditions. Turning to the ASF chain-lengthening characteristics, it is seen that a unique α value of 0.5 for the total $C_n$ is obtained independent of the $H_2$/CO pressure ratio. Because of the strong deviations of short-chain products from the linear ASF plots, only $C_{4+}$ products are considered here. Furthermore, total carbon amounts (resulting from paraffins, terminal olefins and alcohols as well as aldehydes) were calculated for each $C_n$.

The unique $H_2$/CO pressure ratio dependence of the F–T reaction over $Co_4Mn_1K_{0.1}$ resulting in a common ASF behaviour would appear to be in agreement with a single chain-lengthening mechanism for all products, possibly involving a CO insertion step to cause chain initiation and growth. While the rate of the reaction follows a first-order dependence in the hydrogen pressure (Supplementary Fig. 7), no conclusions on the detailed mechanistic pathways can be made this way. On the other hand, since the CO partial pressure in our experiments decreases so as to keep the total pressure constant, CO insertion, if it occurs, would not appear to be a rate-determining step.

The remarkable catalytic performance of CoMnK catalysts is most likely associated with the oxalate route of catalyst preparation, because catalysts with identical or similar composition prepared according to different methods (Supplementary Fig. 8) showed the formation of hydrocarbons (mainly, olefins) but failed to demonstrate their capacity for oxygenates production, which have also been demonstrated in the literature[14–17]. The same oxalate method was previously applied for preparing CoCuMn (ref. 18) and CoCuNb (ref. 20) catalysts. As-prepared CoCuMn was demonstrated to contain core@shell

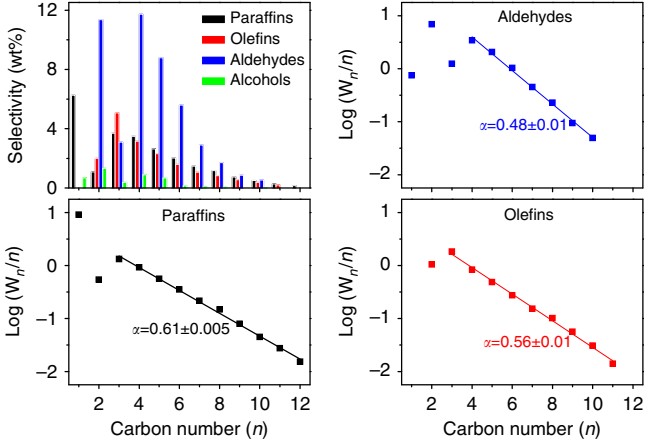

**Figure 2 | Products chain lengthening.** Detailed selectivity patterns and Anderson–Schulz–Flory (ASF) chain-lengthening characteristics of the various product classes over $Co_4Mn_1K_{0.1}$ at 220 °C. An ASF plot for alcohols could not be constructed because of the low fractions of alcohols versus aldehydes.

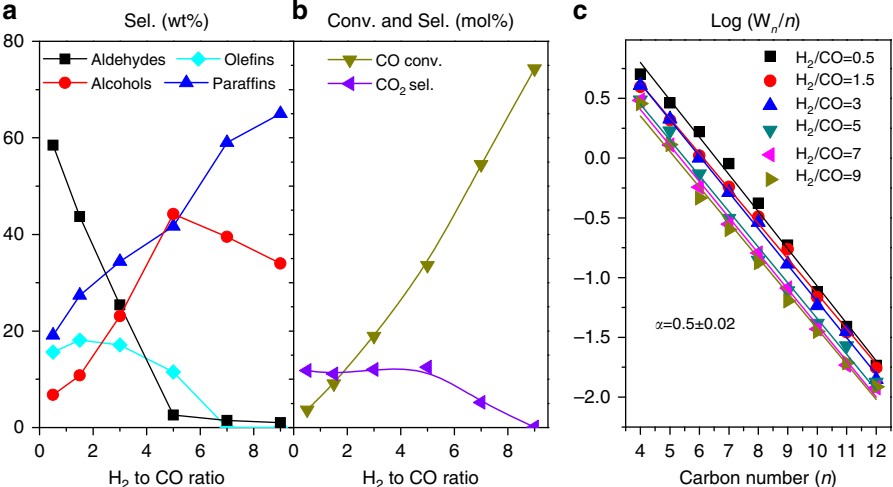

**Figure 3 | Effect of $H_2$ to CO ratio.** (**a**) Products selectivity without $CO_2$, (**b**) CO conversion and $CO_2$ selectivity, (**c**) ASF behaviour for total $C_n$ of the same catalyst showing a chain-lengthening probability independent of the $H_2$/CO pressure ratio. Reaction conditions: $Co_4Mn_1K_{0.1}$ catalyst 0.5 g, 220 °C and 40 bar. The same batch of catalyst was subjected to conversion at different $H_2$/CO ratios (sequential increase from 0.5 to 9 every 24 h). The total time on stream of the catalyst was 144 h. See also Supplementary Fig. 6 for long-term stability tests.

structured nanosized particles besides metal oxide structures. Core@shell structures were identified by atom probe tomography and consisted of a Co dominated core and a thin shell of Cu atoms diluted by Mn and, to a lesser extent, Co. We therefore anticipated similar structures to occur in the case of CoMn, but rather than applying atom probe tomography for a single-grain analysis, we concentrated on revealing the details of the oxidic phases using high-resolution transmission electron microscopy (HRTEM) along with ensemble averaging X-ray powder diffraction (XRD).

**Structural characterization.** Characteristic HRTEM, high-angle annular dark-field imaging (HAADF)-scanning transmission electron microscopy (STEM) structural features and XRD patterns are shown in Fig. 4 for $Co_4Mn_1K_{0.1}$ before and after CO hydrogenation. According to Fig. 4a, metallic Co seems to coexist next to $Mn_5O_8$ (see also Supplementary Fig. 9 for energy-dispersive X-ray chemical mapping and Supplementary Fig. 10 for additional HRTEM images). Fourier transform spectra clearly identify this mixed-valence oxide structure, $Mn_2^{2+}$ $Mn_3^{4+}O_8$, which is best described as consisting of anionic $[Mn^{4+}_3O_8]^{4-}$ planar sheets separated by $Mn^{2+}$ cation layers. It is most probable that the $Mn_5O_8$ phase that forms during the thermal decomposition of the mixed-metal oxalate precursor helps ensure a relatively high specific surface area of $\sim 46\, m^2\, g^{-1}$ (Supplementary Table 2). Besides monoclinic $Mn_5O_8$, metallic Co is also observed in Fig. 4a. The interplanar distance of $\sim 2.05$ Å matches well with the (111) plane of face-centred cubic cobalt metal. According to Supplementary Fig. 11, Co particles and $Mn_5O_8$ seem to jointly form particles with sizes between 11 and 17 nm.

CO hydrogenation has a major influence on the $Co_4Mn_1K_{0.1}$ catalyst bulk composition. As can be seen from HAADF-STEM of $Co_4Mn_1K_{0.1}$ after 24 h time on stream at 240 °C (Fig. 4d–f), despite some significant areal aggregation (Fig. 4e), the $Mn_5O_8$ oxidic phase as present before reaction is retained after reaction (Fig. 4f). However, to our surprise, the metallic Co phase has been completely converted into cobalt carbide after CO hydrogenation. As shown in Fig. 4d, the interplanar distance of $\sim 2.15$ Å can be easily identified as the (111) plane of orthorhombic $Co_2C$. The HAADF-STEM evidence for the occurrence of $Co_2C$ is in full

agreement with the XRD results. As shown in Fig. 4c, metallic cobalt diffractions at $2\theta$ of 44.35° ((111) plane for fcc, JCPDS 15-0806), and 41.7, 44.7 and 47.6° (corresponding to (100), (002) and (101) planes for hcp, JCPDS 05-0727) are identified for the reduced $Co_4Mn_1K_{0.1}$ catalyst. These peaks are absent after catalysis. Major diffraction peaks of $2\theta$ at 41.4, 42.7 and 45.8° correspond well to the (020), (111) and (210) planes of $Co_2C$ (JCPDS 05-0704). Interestingly, the reaction-induced transformation of Co metal into $Co_2C$ does not occur in any significant amounts using potassium-free $Co_4Mn_1$ (Supplementary Fig. 12). We therefore conclude that the formation of bulk $Co_2C$ in the present study is associated with the presence of alkali. Previous studies with iron-based catalysts showed potassium ($K_2O$) to act as a structural stabilizer for high Miller index facets such as Fe (211) and Fe (310) (ref. 21). It has to be noted that the formation of bulk Co carbide during F–T synthesis is rather unusual[22–25]. First, the activation energy for carbon diffusion into cobalt is high (146 kJ mol$^{-1}$ which is about three times higher than for iron)[26]; second, bulk $Co_2C$ has been considered a metastable phase during F–T synthesis; it can easily be hydrogenated to metallic Co and $CH_4$ at relatively low temperature[20,21]. On the other hand, there seems to be mounting evidence from the literature that the formation of surface $Co_2C$ is reaction-induced and that it may be an active phase in the CO hydrogenation over Co-based catalysts[27–32]. It is most probable that synergistic effects at the interface of carbidic $Co_2C$ and oxidic $Mn_5O_8$ are responsible for the remarkable performance of $Co_4Mn_1K_{0.1}$ catalysts in the present study.

**Discussion**

To put into more general perspective the findings of the present study, we have demonstrated that the design of alkali-promoted CoMn catalysts via oxalate precursors leads to unprecedented catalytic performance in terms of straight-chain aldehydes, terminal alcohols and olefins and, more classically, paraffins. The activity and selectivity of the various product classes can be tuned by changing the $H_2$/CO partial pressure ratio at constant total pressure (40 bar). The occurrence of large amounts of terminal oxygenates is most attractive with respect to the possibility of designing a large-scale heterogeneous process. Straight-chain aldehydes and terminal alcohols are being

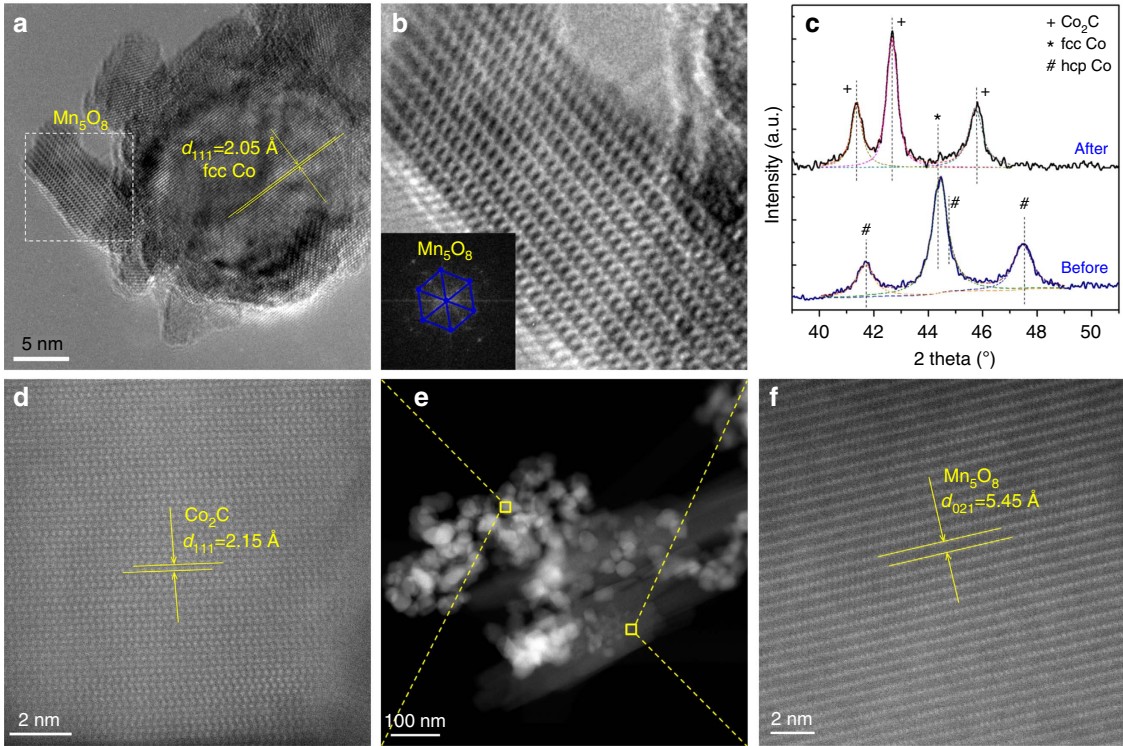

**Figure 4 | Structural characterizations of $Co_4Mn_1K_{0.1}$ catalysts.** (**a**) HRTEM image of $Co_4Mn_1K_{0.1}$ catalyst before reaction; (**b**) enlarged HRTEM image and (inset) its Fourier transform of the selected region of **a**; (**c**) XRD patterns of $Co_4Mn_1K_{0.1}$ catalyst: both before and after CO hydrogenation; (**d**–**f**) HAADF-STEM images of $Co_4Mn_1K_{0.1}$ catalyst after CO hydrogenation.

extensively used as feedstocks for plasticizers, detergents, lubricants, food additives or perfumes, dependent on the chain length. More specifically, short-chain alcohols are important fuel additives. $C_{3+}$ aldehydes are currently produced via the homogeneous 'oxo' process (hydroformylation) of a terminal $C_{2+}$ olefin in the presence of a mixture of CO and $H_2$ at high pressure in the liquid phase. Subsequent hydrogenation leads to the respective $C_{3+}$ alcohol. Recently, tandem hydroformylation/ hydrogenation in a one-pot approach with a single homogeneous noble metal catalyst demonstrated simultaneous hydroformylation and chemoselective reduction with hydrogen of the intermediate aldehyde to alcohol[33]. While the hydroformylation process produces iso-aldehydes as by-product (imposed by the Markovnikov rule), strictly linear $C_n$ aldehydes and alcohols are formed from CO and $H_2$ (in the absence of a starting olefin) using the heterogeneous one-pot F–T technology. The chain length of these oxygenates is dictated by the ASF distribution and common unit operations of chemical engineering allow a straightforward sorting according to $C_n$ products with unique $n$.

The 'oxo' process was initially discovered by Roelen in 1938 (ref. 34) when he recycled gasol (light hydrocarbons, including olefins and, in particular, ethylene) to improve the performance of the heterogeneous F–T reaction to paraffins. However, it escaped his attention that aldehyde formation under high-pressure conditions was actually a process of homogeneous rather than heterogeneous CO hydrogenation, as shown later by Heck and Breslow[35]. Oxygenate formation under our pressure conditions is clearly heterogeneous since neither catalyst loss nor iso-product formation were observed. Similar conclusions were drawn by Quek *et al.* from batch-type studies in the liquid phase using Ru nanoparticles at pressures similar to ours[9].

The physico-chemical characterization of our potassium-promoted CoMn catalysts revealed the occurrence of a potassium-stabilized $Co_2C$ (after reaction) along with a mixed-valence

$Mn_2(II)Mn_3(IV)O_8$ phase (both before and after reaction). Both phases seem to be in intimate contact during the synthesis and are anticipated to act synergistically to produce straight-chain oxygenates (*n*-aldehydes and terminal *n*-alcohols), α-olefins and *n*-paraffins with selectivities depending strongly on the $H_2$/CO partial pressure. Summing $C_{4+}$ oxygenates and hydrocarbons for each $C_n$, a chain-lengthening probability independent of the $H_2$ partial pressure was obtained. This would appear to be in agreement with a unique chain-lengthening mechanism for all product classes. A definitive proof yet cannot be provided on the basis of steady-state kinetic experiments but would need extensive spectroscopic and microscopic in operando investigations as presently performed in a number of laboratories.

## Methods

**Oxalate precipitation.** CoMnK oxalates were precipitated according to the following recipe. First, a mixed solution of $Co(NO_3)_2·6H_2O$ and $Mn(NO_3)_2·4H_2O$ in acetone (100 ml, ~0.5 mol l$^{-1}$), an aqueous solution of $KNO_3$ (5 ml, ~0.2 mol l$^{-1}$), and a solution of $H_2C_2O_4·2H_2O$ in acetone (150 ml, ~0.5 mol l$^{-1}$) were prepared in three separate beakers. Then, the mixed acetonic solution of $Co(NO_3)_2·6H_2O$ and $Mn(NO_3)_2·4H_2O$, and the aqueous solution of $KNO_3$ were added fast and simultaneously to the acetonic solution of $H_2C_2O_4·2H_2O$ under vigorous stirring. Stirring was kept for at least 5 min until the colour of the precipitates appeared homogeneous. The atomic ratios of Co/Mn were kept at 2/1 and 4/1, and the ratios between K/Mn were 0.1/1 and 0.3/1, respectively. Then, the slurries were kept overnight. After removal of the supernatant acetone, the precipitate was centrifuged, dried overnight at 110 °C, and finally crushed and sieved so as to obtain a size fraction between 125 and 250 μm for characterization and high-pressure catalytic investigations. The co-precipitation of $K_2C_2O_4$ as described above was enforced by a solubility/entrainment effect since $KNO_3$ is soluble in water, but not in acetone. It is also important to mention that the supernatant solutions after precipitation were clear and transparent. See Supplementary Methods for the preparation of binary CoMn oxalate and characterization methods.

**Catalytic tests.** High-pressure catalytic tests were performed in a fixed-bed plug-flow reactor consisting of a quartz tubule ($\Phi_{inner} = 7$ mm) inserted into a stainless steel housing. A condenser along with a gas–liquid separator was mounted

at the reactor outlet. Typically, 1.3 g of oxalate was diluted with up to 2 g of SiC to achieve isothermal plug-flow conditions followed by *in situ* TPDec in $H_2$ at 0.1 MPa (30 ml min$^{-1}$) and 370 °C for 1 h (after oxalate decomposition, the amount of activated catalyst is ~0.5 g). The reactor was subsequently cooled to a temperature below 100 °C in flowing hydrogen before adding CO so as to produce a syngas feed with the desired $H_2$/CO ratio. Metal carbonyls (mainly, Ni(CO)$_4$) were removed by passing the CO feed through a zeolite 4A trap at high temperature before introduction into the reactor. Typical flow rates were $H_2$/CO = 24/16 ml min$^{-1}$ providing a GHSV = 3,600 h$^{-1}$ (gas hourly space velocity). After pressurizing the system to 40 bar, the temperature was raised using low heating rates of 1 °C min$^{-1}$ up to 200 °C and kept overnight under these conditions. Catalytic activities and product selectivities were determined after stabilization for at least 12 h. The CO conversion and product selectivities were measured by online GC–MS (Agilent 7890A GC/5975 MS).

**Data availability.** Experimental data supporting the findings of this investigation are available from the corresponding authors on reasonable request.

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

## Acknowledgements

The project has been gratefully supported by National Science Foundation under contract no. CBET-1438227. We sincerely thank Professor Xiaonian Li (Industrial Catalysis Institute of Zhejiang University of Technology) for assisting with HRTEM measurements and Dr Libor Kovarik (Pacific Northwest National Laboratory) for helping with HAADF-STEM measurements.

## Author contributions

Y.X. performed all the measurements and contributed to drafting the paper. N.K. initiated the research project and drafted the paper.

## Additional information

**Competing financial interests:** The authors declare no competing financial interests.

**How to cite this article**: Xiang, Y & Kruse, N. Tuning the catalytic CO hydrogenation to straight- and long-chain aldehydes/alcohols and olefins/paraffins. *Nat. Commun.* **7**, 13058 doi: 10.1038/ncomms13058 (2016).

