## [Peer review file · Nature Communications]

Reviewers' comments:

Reviewer #1 (Remarks to the Author):

The authors are describing results to show the production of oxygenates from syngas using a rather complex catalyst. The results are interesting and should be of interest. The experimental approach is valid and should produce high quality results.

Most of the data are for low conversions where oxygenates can be formed. It would be helpful if the authors indicate whether the data in Figure 3 are typical; if so, then their results are not significantly different from those of simpler catalysts. The authors should indicate through the manuscript where they are considering low CO conversion data. Their own data in Figure 3 clearly show that their catalyst for the conditions used is not selective for oxygenates at conditions that would be used in an industrial process.

The peaks for Co hpc at 41.8 and 47.8 in Figure 4 look different from the literature. It would be helpful if the authors could consider this point and provide some discussion if indeed they are different.

It is surprising that cobalt carbide is formed under the FT conditions since most find that cobalt carbide disappears under FT reaction condition.

The authors should indicate whether a fresh batch of catalyst was used at each temperature or if the same batch was subjected to conversion at various temperatures.

It would be helpful if the authors could provide some indication of catalyst stability.

The references appear to be appropriate.

The manuscript is well written.

Reviewer #2 (Remarks to the Author):

The manuscript of Prof. Kruse reported the production of straight-chain aldehydes, alcohols, terminal olefins, and paraffins on K-promoted CoMn catalysts, which are prepared by the oxalate coprecipitation method, in the hydrogenation of CO (Fischer-Tropsch synthesis) by varying the H₂/CO ratio. They found that the products contain about 60% n-aldehydes at a H₂/CO ratio of 0.5, and a 65%/35% paraffin/alcohol ratio is obtained at a ratio of 9. But the H₂/CO ratio does not affect the ASF distribution, and the products obey the ASF distribution except for the short-chain products (mainly below C₄). Interestingly, a bulk Co₂C phase was identified after reaction. Its synergistic interaction with Mn₅O₈ is proposed to be responsible for the unique product distribution. The finding is in general interesting and may be beneficial for the development of a versatile FTS technology. However, it is apparent that more information should be provided and clarified.

(1) Catalyst preparation. The concentrations of the precursors should be provided.

(2) Catalytic testing. Has the activity testing been reproduced? How about the carbon balance?

(3) There are some typos to be corrected. For example, line 17, "oxidic" is not a noun. Line 41, "dependence in" should be "dependence on". Line 96, "(" is missing.

(4) More HRTEM images are needed to substantiate the microstructures of the Co₄Mn₁K_{0.1} catalyst before and after reaction. Are the HRTEM images presented are typical? The presence of hcp Co should be indicated on the HRTEM image of the catalyst before reaction. Where is the magnesium oxide on the used catalyst? Has it changed to phase other than Mn₅O₈ after reaction? Is it still present as a shell for Co₂C?

(5) The long-term stability testing of at least the Co₄Mn₁K_{0.1} catalyst with the evolutions of CO and the products is needed. This experiment, along with the monitoring of the phase change during the reaction, may clarify whether the active phase for the formation of oxygenates is really Co₂C or not.

(6) Figure S7. It is unclear which species the PSD histogram belongs to and how the authors constructed it.

Reviewer #3 (Remarks to the Author):

An interesting study of the effect of the H₂/CO ratio on the selectivity of CO hydrogenation over CoMn(K) catalysts is presented. While some of the reported effects are well-known in the literature, the comprehensive nature of the current study is certainly an important addition. Also the historical background in the introduction and in the concluding remarks lift this paper well above the general literature. The independence of the chain growth probability of the H₂/CO ratio, even though the rate varies dramatically when this ratio is changed, is remarkable and warrants further studies.

The suggested drawbacks of the homogeneous hydroformylation process are overstated. Industrial practice shows that hydroformylation operates with sub-ppm quantities of Rh, and Rh stays in the reactor for several years - it seems unlikely that the current CoMnK catalyst will replace this technology.

The most puzzling part of the manuscript is the structural interpretation. The authors report the formation of Co₂C after reaction over CoMnK catalysts, but not over CoMn. They attribute the high oxygenate selectivity to the formation of a bulk Co₂C phase. Since they have exposed their catalyst to low H₂/CO ratios, the formation of bulk Co₂C is not unexpected. Is this phase however also observed when the catalyst is only exposed to typical H₂/CO ratios?

Co₂C should be unstable under CO hydrogenation conditions. Did the authors try to hydrogenate the CoMnK catalyst after reaction to test the stability of the Co₂C phase? Do they observe easy decomposition to CH₄ and metallic Co? This would suggest this phase is not present under typical reaction conditions.

It seems unlikely that the Co₂C phase is responsible for the unique selectivity over the range of conditions. Is Co₂C also formed for higher H₂/CO ratios? The CoMn catalyst does not form Co₂C, yet it also produces substantial amounts of oxygenates.

The comparison between CoMnK prepared via the oxalate route and the conventional route (line 162-165) should not be made since the conventional catalysts were not tested under identical conditions. The suggestion that the Co₂C formation is related to the catalyst preparation method is also not demonstrated since no comparison was made between preparation methods. It is however clear that K is required to form/stabilize the Co₂C phase.

Reviewer #1 (Remarks to the Author):

The authors are describing results to show the production of oxygenates from syngas using a rather complex catalyst. The results are interesting and should be of interest. The experimental approach is valid and should produce high quality results.

Q1: Most of the data are for low conversions where oxygenates can be formed. It would be helpful if the authors indicate whether the data in Figure 3 are typical; if so, then their results are not significantly different from those of simpler catalysts. The authors should indicate through the manuscript where they are considering low CO conversion data. Their own data in Figure 3 clearly show that their catalyst for the conditions used is not selective for oxygenates at conditions that would be used in an industrial process.

Reply: It is well known that the formation of oxygenates through the Fischer-Tropsch synthesis is thermodynamically unfavorable at high temperature where high CO conversion can be obtained.

So, the reviewer is absolutely right that high oxygenates selectivity (especially n-aldehydes) is usually obtained for low CO conversion (Figure 1). However, the data in Figure 3 shows that (relatively) high alcohols selectivity (35 wt%) can also be obtained at high CO conversion (~75%) when employing a H₂/CO partial pressure ratio of 9/1. Such results are typical for CoMnK catalysts (in particular Co₄Mn₁K_{0.1}) and we are not aware, to the best of our knowledge, that such behavior was reported any time before for whatever catalyst composition. We also studied the influence of the H₂/CO pressure ratio for other catalysts such as CoMnTi, CoCu and CoCuK, however, selectivity patterns similar to those shown in Figure 3 were not observed.

It should be noted that the oxygenates selectivities have always been calculated by taking into account the CO₂ production. This is part of the reason for the decreasing trend with increasing temperatures (or higher CO conversion). In industrial applications, FT product selectivities are frequently calculated without considering CO₂ formation, for example, when H₂-lean syngas (from coal or biomass) is employed and the water-gas-shift reaction has to be used to adjust the H₂/CO ratio during the FT reaction. Selectivity calculations ex-CO₂ can also be found throughout the FT academic literature (to our opinion frequently inappropriate). If this is done for measurements as reported in the present paper oxygenates selectivities up to ~50 wt% can be obtained at a CO conversion of more than 40% (see figure below, right panel)

Figure S1. Total oxygenates selectivity (without considering CO₂ formation) as a function of reaction temperature and CO conversion.

The above two Figures have been added to the supporting information (Figure S1). The following sentences have been altered/added to the main text (~ line 77-79).

Oxygenates form with highest selectivities (calculations always include CO₂ production) under conditions of low CO conversion (low temperature). If the CO₂ production in our selectivity calculations is ignored (this procedure can be frequently found in the FT literature with H₂-lean syngas), oxygenates selectivities up to ~50 wt% can be obtained for a CO conversion of ~40 % at high temperature (see Figure S1).

Q2: The peaks for Co hpc at 41.8 and 47.8 in Figure 4 look different from the literature. It would be helpful if the authors could consider this point and provide some discussion if indeed they are different.

Reply: We double-checked our XRD results and found they can be consistently understood as being due to hcp cobalt. The diffraction peaks at 41.7° , 44.7° and 47.6° correspond well to the crystal planes of (100), (002) and (101), respectively, according to JCPDS card no 05-0727. The same data is being reported in the literature (Barakat et al. *Nanoscale Res Lett* 2014 3; 9(1))

We state now more precisely in the text:

As shown in Figure 4D, metallic cobalt diffractions at 2θ of 44.35° ((111) plane for fcc, JCPDS 15-0806), and 41.7° , 44.7° and 47.6° (corresponding to (100), (002) and (101) planes for hcp, JCPDS 05-0727) are identified for the reduced $\text{Co}_4\text{Mn}_1\text{K}_{0.1}$ catalyst.

Q3: It is surprising that cobalt carbide is formed under the FT conditions since most find that cobalt carbide disappears under FT reaction condition.

Reply: The formation of cobalt carbide under the FT conditions is indeed surprising. So, we completely agree with the referee's view. The elder literature (published in 1940s by Weller) and more recently the work by the Burtron Davis group all found that cobalt carbide disappears under FT reaction conditions. Our XRD results were, however, reproduced at least three times. They all demonstrated bulk Co_2C was formed. Additionally, we compared the XRD patterns of the Co_4Mn_1 and $\text{Co}_4\text{Mn}_1\text{K}_{0.1}$ catalysts after FT synthesis (see the Figure below) and found that bulk Co_2C was only observed for $\text{Co}_4\text{Mn}_1\text{K}_{0.1}$ catalyst whilst only metallic Co (fcc) was observed for Co_4Mn_1 . Additionally, our more recent HRTEM and STEM-EELS studies at PNNL (Pacific Northwest National Laboratories) confirmed the formation of Co_2C . A detailed account of Co_2C formation will be given in a forthcoming paper.

A new Figure including the XRD patterns of Co_4Mn_1 , $\text{Co}_4\text{Mn}_1\text{Li}_{0.1}$, $\text{Co}_4\text{Mn}_1\text{Na}_{0.1}$ and $\text{Co}_4\text{Mn}_1\text{K}_{0.1}$ catalysts after FT synthesis has been added to the Supporting Information, Figure S12.

Figure 2. XRD patterns of the Co_4Mn_1 and $\text{Co}_4\text{Mn}_1\text{K}_{0.1}$ catalysts after FT synthesis.

Q4: The authors should indicate whether a fresh batch of catalyst was used at each temperature or if the same batch was subjected to conversion at various temperatures.

Reply: The same batch of catalyst was subjected to conversion at various temperatures (Figure 1) and H₂/CO ratio (Figure 3).

We have added this to the caption of Figure 1 and Figure 3.

Q4: It would be helpful if the authors could provide some indication of catalyst stability.

Reply: The stability test has been performed on Co₄Mn₁K_{0.1} catalyst at 220 °C, 40 bar, H₂/CO=1.5 for 100 h. The activity (CO conversion) and total oxygenates selectivity (without CO₂) as a function of time-on-stream has been added to the Supporting Information (Figure S6).

Q5: The references appear to be appropriate. The manuscript is well written.

Reply: Thank you!

Reviewer #2 (Remarks to the Author):

The manuscript of Prof. Kruse reported the production of straight-chain aldehydes, alcohols, terminal olefins, and paraffins on K-promoted CoMn catalysts, which are prepared by the oxalate coprecipitation method, in the hydrogenation of CO (Fischer-Tropsch synthesis) by varying the H₂/CO ratio. They found that the products contain about 60% n-aldehydes at a H₂/CO ratio of 0.5, and a 65%/35% paraffin/alcohol ratio is obtained at a ratio of 9. But the H₂/CO ratio does not affect the ASF distribution, and the products obey the ASF distribution except for the short-chain products (mainly below C₄). Interestingly, a bulk Co₂C phase was identified after reaction. Its synergistic interaction with Mn₅O₈ is proposed to be responsible for the unique product distribution. The finding is in general interesting and may be beneficial for the development of a versatile FTS technology. However, it is apparent that more information should be provided and clarified.

Q1: Catalyst preparation. The concentrations of the precursors should be provided.

Reply: We have added the concentrations of the precursors to the Methods.

Q2: Catalytic testing. Has the activity testing been reproduced? How about the carbon balance?

Reply: We reproduced the catalytic performance of Co₄Mn₁K_{0.1} at least 5 times over different batches of catalyst, the error in terms of activity and selectivity was less than 10%.

The carbon balance has been calculated through comparison between the CO conversion (using an external standard) and the total carbon as detected in the products. The error was ~8% from three different measurements.

We have added the following sentence to the caption of Figure 1.

The catalytic performance of Co₄Mn₁K_{0.1} at 220 °C was reproduced at least five times with errors less than 10% over different batches of catalyst. The error for the total carbon balance was ~8% from three different measurements.

Q3: There are some typos to be corrected. For example, line 17, "oxidic" is not a noun. Line 41, "dependence in" should be "dependence on". Line 96, "(" is missing.

Reply: We have corrected the typos.

Q4: More HRTEM images are needed to substantiate the microstructures of the $\text{Co}_4\text{Mn}_1\text{K}_{0.1}$ catalyst before and after reaction. Are the HRTEM images presented typical? The presence of hcp Co should be indicated on the HRTEM image of the catalyst before reaction. Where is the magnesium oxide on the used catalyst? Has it changed to phase other than Mn_5O_8 after reaction? Is it still present as a shell for Co_2C ?

Reply: Yes, the HRTEM data is representative. Additional HRTEM images for the $\text{Co}_4\text{Mn}_1\text{K}_{0.1}$ catalyst have been added to the supporting information (Figure S10). Before reaction, both hcp and fcc cobalt were identified through FFT. More recently, we did additional detailed characterization studies using HRTEM and high resolution STEM-EELS at PNNL (Pacific Northwest National Laboratories). The results presented in this paper have been reproduced and additional evidence obtained.

For the $\text{Co}_4\text{Mn}_1\text{K}_{0.1}$ catalyst after reaction, using STEM-EDX (Figure S9), a homogeneous distribution of all three elements was observed both before and after reaction. In some specific regions (between two particles), a separate Mn phase was also observed.

According to the recent studies at PNNL, significant amounts of manganese oxide appeared aggregated in the catalysts after reaction. However, such aggregation did not affect the chemical nature of the oxide phase: the Mn_5O_8 structure as identified before the reaction studies was retained after them. Additionally, a core@shell type structure was observed, with Co_2C forming the core while Mn and Co were both present in the shell. However, we think the detailed characterization results are out of the scope of the present paper, a follow-up paper is currently in preparation.

Q5: The long-term stability testing of at least the $\text{Co}_4\text{Mn}_1\text{K}_{0.1}$ catalyst with the evolutions of CO and the products is needed. This experiment, along with the monitoring of the phase change during the reaction, may clarify whether the active phase for the formation of oxygenates is really Co_2C or not.

Reply: The long-term stability testing has been performed on $\text{Co}_4\text{Mn}_1\text{K}_{0.1}$ catalyst at 220°C , 40 bar, $\text{H}_2/\text{CO}=1.5$ for 100 h. The activity (CO conversion) and total oxygenates selectivity (without CO_2) as a function of time-on-stream has been added to the Supporting Information (Figure S6).

We agree with the referee that the in-situ monitoring of the phase change during the on-going reaction would be most interesting. Such experiments are, however, rather challenging and difficult to accomplish for several reasons. First, the catalyst has to be diluted with SiC and second, the pressure for the reaction is high. On the other hand, detailed ex-situ characterization studies of the catalyst were performed after the reaction demonstrating the importance of Co_2C for the observed catalytic performance. A follow-up paper is in preparation.

Q6: Figure S7. It is unclear which species the PSD histogram belongs to and how the authors constructed it.

Reply: We have added the explanation of particle size distribution to the caption of Figure S7. The following sentence has been added:

More than 200 particles of the shown image as well as of other images were evaluated for their size so as to determine the percentage of each particle size range. These particles are composed of mainly cobalt with also certain amounts of manganese according to STEM-EDX (Figure S9).

Reviewer #3 (Remarks to the Author):

An interesting study of the effect of the H₂/CO ratio on the selectivity of CO hydrogenation over CoMn(K) catalysts is presented. While some of the reported effects are well-known in the literature, the comprehensive nature of the current study is certainly an important addition. Also the historical background in the introduction and in the concluding remarks lift this paper well above the general literature. The independence of the chain growth probability of the H₂/CO ratio, even though the rate varies dramatically when this ratio is changed, is remarkable and warrants further studies.

Q1: The suggested drawbacks of the homogeneous hydroformylation process are overstated. Industrial practice shows that hydroformylation operates with sub-ppm quantities of Rh, and Rh stays in the reactor for several years - it seems unlikely that the current CoMnK catalyst will replace this technology.

Reply: We agree with the referee that important progress has been made over the years with homogeneous hydroformylation. Rh slip in the recovery is much less of a problem than decades ago. Progress has also been made with respect to the regioselectivity, as mentioned in the text.

We agree that CoMnK catalysts will still have to demonstrate their usefulness as a potential candidate for heterogeneous hydroformylation. [redacted]

The message of the present paper was to demonstrate an alternative method for n-aldehydes production exists through heterogeneous catalysis and to show how the product distribution can be tuned. Rather than using petro-olefins and syngas as the starting material for hydroformylation, the FT synthesis uses only syngas (derived from various resources), in the absence of a target olefin.

Nevertheless, we took into account the comment of the referee and changed the sentence “A critical issue related with hydroformylation yet remains the recovery of the noble metal catalyst (usually a homonuclear Rh-based complex) and the limited regioselectivity imposed by the Markovnikov rule. Using the heterogeneous “one-pot” F-T technology, strictly linear C_n aldehydes and alcohols are produced from CO and H₂ (in the absence of a starting olefin).”

to “While the hydroformylation process produces iso-aldehydes as by-product (imposed by the Markovnikov rule), strictly linear C_n aldehydes and alcohols are formed from CO and H₂ (in the absence of a starting olefin) using the heterogeneous “one-pot” F-T technology.”

Q2: The most puzzling part of the manuscript is the structural interpretation. The authors report the formation of Co_2C after reaction over CoMnK catalysts, but not over CoMn. They attribute the high oxygenate selectivity to the formation of a bulk Co_2C phase. Since they have exposed their catalyst to low H_2/CO ratios, the formation of bulk Co_2C is not unexpected. Is this phase however also observed when the catalyst is only exposed to typical H_2/CO ratios?

Reply: The formation of bulk Co_2C is actually a very interesting result of the characterization part of our studies. Both XRD and HRTEM characterization clearly indicate the formation of Co_2C after FT reaction (for a H_2/CO ratio at and below 1.5). The formation of Co_2C was confirmed in more recent HRTEM and STEM-EELS studies (to be published in a forthcoming paper). Furthermore, we observed that once the Co_2C was formed in studies with low H_2/CO ratio of 1.5 or 1, it remains stable at higher H_2/CO ratios (see answer to Q3). On the other hand, we found that if a fresh catalyst was exposed directly to syngas with high H_2/CO ratio (5/1 or 7/1), the transformation into a carbide was not observed. It is then interesting to note that major hysteresis effects in the catalytic performance can be observed. These hysteresis effects seem to correlate with the occurrence of a bulk carbide.

Q3: Co_2C should be unstable under CO hydrogenation conditions. Did the authors try to hydrogenate the CoMnK catalyst after reaction to test the stability of the Co_2C phase? Do they observe easy decomposition to CH_4 and metallic Co? This would suggest this phase is not present under typical reaction conditions.

Reply: We have mentioned in the original manuscript, “*the formation of bulk Co carbide during FT synthesis is rather unusual. First, the activation energy for carbon diffusion into cobalt is high (146 kJ/mol which is about 3 times higher than for iron); Second, bulk Co_2C has been considered a metastable species during FT synthesis; it can be easily hydrogenated to metallic Co and CH_4 at relatively low temperature.*”

We do agree with the referee that the Co_2C phase is most probably unstable under pure H_2 at high temperature. However, from our more recent results, we can conclude that the Co_2C phase, once it has been formed, is stable under syngas (even at high H_2/CO ratio) conditions. As shown in the following Figure, the Co_2C phase (pre-formed at low H_2/CO ratio) was retained after reaction at high H_2/CO ratio (7/1).

We plan to perform H_2 -TPR experiments of the used catalysts in conjunction with XRD studies in the near future so as to address the question for the reducibility of the Co_2C phase.

[redacted]

Q4: It seems unlikely that the Co_2C phase is responsible for the unique selectivity over the range of conditions. Is Co_2C also formed for higher H_2/CO ratios? The CoMn catalyst does not form Co_2C , yet it also produces substantial amounts of oxygenates.

Reply: As shown in the above Figure 3, Co_2C does not form directly at high H_2/CO ratios (such as 5/1 and 7/1), but once the Co_2C was pre-formed at low H_2/CO ratio 1.5/1 conditions, it remains stable at higher H_2/CO ratios. The catalytic results for a $\text{Co}_4\text{Mn}_1\text{K}_{0.1}$ catalyst subjected to syngas at $\text{H}_2/\text{CO}=5$ and 7, without preconditioning at low H_2/CO ratios, shows very low CO conversion

with formation of only paraffins (mainly CH₄). We have rather detailed results about this run-in effect as well as the respective characterization results; a follow-up paper describing these results is in preparation.

Bulk Co₂C does not form over CoMn according to XRD. However, a surface carbidization cannot be excluded. More detailed surface characterization of the catalysts after reaction has begun in our laboratory. Indeed, a tiny peak representative of Co₂C was also observed in XRD of Co₄Mn₁ after reaction (see the following Figure). We recognize that environmental XPS studies would be most suitable to tackle the problem. Such measurements are also planned.

XRD patterns of the Co₄Mn₁ and Co₄Mn₁K_{0.1} catalysts after FT synthesis.

Q5: The comparison between CoMnK prepared via the oxalate route and the conventional route (line 162-165) should not be made since the conventional catalysts were not tested under identical conditions. The suggestion that the Co₂C formation is related to the catalyst preparation method is also not demonstrated since no comparison was made between preparation methods. It is however clear that K is required to form/stabilize the Co₂C phase.

Reply: We actually compared the catalytic performance of CoMn prepared via the oxalate route with that of Co nanoparticles (prepared from hot-injection of Co₂(CO)₈, see reference [Iablokov et al. Nano Letters, 2012, 12: 3091-3096]) supported on MnO_x (x = ~1.6, close to Mn₅O₈, prepared by thermal decomposition of Mn-oxalate). The results have been added to the Supporting Information (Figure S8). It is clearly seen that the total oxygenates selectivity for Co₂Mn₁ (via oxalate co-precipitation) is significantly higher than for Co/MnO_x (impregnation with Co nanoparticles onto MnO_x).

Figure S8 Comparison of the selectivity between a traditional, supported Co/MnO_x catalyst (prepared by impregnation) and an unsupported CoMn catalyst prepared through oxalate co-precipitation. The MnO_x support was prepared by thermal decomposition of Mn-oxalate. The Co particle size was around 10 nm for Co/MnO_x.

We changed the original sentence “the formation of bulk Co₂C in the present study is associated with the catalyst preparation method and the presence of alkali.”

to “the formation of bulk Co₂C in the present study is associated with the presence of alkali.”

Reviewers' comments:

Reviewer #1 (Remarks to the Author):

The authors have attempted to respond to the reviewers comments and it is suggested that the manuscript be accepted.

Reviewer #2 (Remarks to the Author):

The revised manuscript addressed all my comments. However, the replies to the fourth and the fifth comments are not satisfactory. The authors are encouraged to show the new results of the catalyst after reaction from PNNL in the present manuscript, as from the present results the microstructure of the catalyst after reaction still remains unclear. The long term testing is also not convincing, as the conversion is too low to testify its stability. The stability testing at higher conversion is necessary. And the XRD result after stability testing at higher conversion will be helpful to a better understanding of the nature of the active phase.

Reviewer #3 (Remarks to the Author):

The authors have addressed my comments adequately. The presence and long term stability of the Co₂C phase remain surprising.

Reviewer #1 (Remarks to the Author):

The authors have attempted to respond to the reviewers comments and it is suggested that the manuscript be accepted.

Reply: Thank you!

Reviewer #2 (Remarks to the Author):

The revised manuscript addressed all my comments. However, the replies to the fourth and the fifth comments are not satisfactory. The authors are encouraged to show the new results of the catalyst after reaction from PNNL in the present manuscript, as from the present results the microstructure of the catalyst after reaction still remains unclear. The long term testing is also not convincing, as the conversion is too low to testify its stability. The stability testing at higher conversion is necessary. And the XRD result after stability testing at higher conversion will be helpful to a better understanding of the nature of the active phase.

Reply: 1. The following microscopy results have been added to Figure 4.

The following sentences have been added to the main text:

As can be seen from HAADF-STEM of $\text{Co}_4\text{Mn}_1\text{K}_{0.1}$ after 24 h time-on-stream at 240°C (Fig. 4D-F), despite some significant areal aggregation (Fig. 4E), the Mn_2O_8 oxidic phase as present before reaction is retained after reaction (Fig. 4F). However, to our surprise, the metallic Co phase has been completely converted into cobalt carbide after CO hydrogenation. As shown in Fig. 4D, the interplanar distance of $\sim 2.15\text{\AA}$ can be easily identified as the (111) plane of orthorhombic Co_2C . The HAADF-STEM evidence for the occurrence of Co_2C is in full agreement with the XRD results.

2. Long term stability testing has been performed at 260°C with relatively high CO conversion ($\sim 40\%$). The following new figure has been added to Fig. S6.

The following sentences have been added to the caption of Fig. S6

(A) and (B): Activity (CO conversion) and total oxygenates selectivity (without CO₂) as a function of time-on-stream for Co₄Mn₁K_{0.1} catalyst at 220°C (Panel A) and 260°C (Panel B), 40 bar, H₂/CO=1.5.

It is seen that at both temperatures, the selectivity of total oxygenates (without CO₂) is remarkably stable within the investigated ranges of time-on-stream (between 56 to 62 wt% at 220°C and 40 to 47 wt% at 260°C, respectively). The selectivity of hydrocarbons (sum of paraffins and olefins) and CO₂ are also stable. At 220°C, the CO conversion during the measurements was always between 5 and 10%, but never below 5% within 112 h time-on-stream. At 260°C, the CO conversion slightly decreased from 44 to 35% after 96 h time-on-stream, however, the total oxygenates yield (CO conversion * oxygenates selectivity) remained strikingly stable (between 15 to 17 wt%).

3. XRD characterization was performed for Co₄Mn₁K_{0.1} after stability tests at 260°C for 96 h (CO conversion > 35%). The following sentences have been added to the caption along with Fig. S6 (C).

XRD pattern of Co₄Mn₁K_{0.1} after long-term stability testing for 96 h at 260°C. The XRD measurement was performed with the SiC-diluted catalyst as recovered from the reactor. The diffraction spectrum at 2 theta between 39-50° (typical diffraction region for Co metal and Co carbides) shows clearly the occurrence of a Co₂C phase while the Co metallic phase as initially present in the virgin catalyst, is completely absent.

Reviewer #3 (Remarks to the Author):

The authors have addressed my comments adequately. The presence and long term stability of the Co₂C phase remain surprising.

Reply: We agree. See above for the added XRD pattern after long-term reaction studies.

REVIEWERS' COMMENTS:

Reviewer #2 (Remarks to the Author):

The authors have addressed my concerns, so the manuscript is now acceptable.

REVIEWERS' COMMENTS:

Reviewer #2 (Remarks to the Author):

The authors have addressed my concerns, so the manuscript is now acceptable.

Reply: Thank you!